# DOPAnization of tyrosine in α-synuclein by tyrosine hydroxylase leads to the formation of oligomers

Mingyue Jin [1,2,8], Sakiko Matsumoto [1,8], Takashi Ayaki[3,8], Hodaka Yamakado[3], Tomoyuki Taguchi[3], Natsuko Togawa [3], Ayumu Konno [4], Hirokazu Hirai [4], Hiroshi Nakajima[5], Shoji Komai[6], Ryuichi Ishida[1], Syuhei Chiba[1], Ryosuke Takahashi [3], Toshifumi Takao[7] & Shinji Hirotsune [1] ✉

Parkinson's disease is a progressive neurodegenerative disorder characterized by the preferential loss of tyrosine hydroxylase (TH)-expressing dopaminergic neurons in the substantia nigra. Although the abnormal accumulation and aggregation of α-synuclein have been implicated in the pathogenesis of Parkinson's disease, the underlying mechanisms remain largely elusive. Here, we found that TH converts Tyr136 in α-synuclein into dihydroxyphenylalanine (DOPA; Y136DOPA) through mass spectrometric analysis. Y136DOPA modification was clearly detected by a specific antibody in the dopaminergic neurons of α-synuclein-overexpressing mice as well as human α-synucleinopathies. Furthermore, dopanized α-synuclein tended to form oligomers rather than large fibril aggregates and significantly enhanced neurotoxicity. Our findings suggest that the dopanization of α-synuclein by TH may contribute to oligomer and/or seed formation causing neurodegeneration with the potential to shed light on the pathogenesis of Parkinson's disease.

Altered motor symptoms caused by Parkinson's disease (PD) are mainly attributable to the preferential loss of dopaminergic neurons in the substantia nigra (SN)[1,2]. The α-synuclein (αSyn) gene is considered a major genetic contributor of this disease. Several point mutations, such as E46K or A53T, and genomic multiplications in the αSyn gene have been linked to familial PD[3–7]. αSyn is a natively unfolded protein constituted by 140 amino acids with three distinct regions including an amphipathic N-terminal region (1–60), a hydrophobic nonamyloid-β component region (61–95) and a highly acidic C-terminal region (96–140)[8]. It is physiologically involved in neurotransmitter release at presynaptic terminals and in microtubule

regulation as a microtubule-associated protein (MAP)[9–11]. In PD brain, αSyn forms an abnormal aggregation that is the major component of Lewy pathology[12]. This aggregation has a conformational polymorphism that varies from a small oligomer to an amyloid fibril, which generates biochemical differences in neurotoxicity[13,14]. In particular, recent studies indicate that the αSyn oligomeric species induce more pathogenic effects and play a key role in PD[15–17]. Interestingly, even though αSyn is abundantly expressed in neurons throughout the entire brain, PD pathology displays the clear tropism of neurodegeneration in dopaminergic neurons, such as the SN neurons.

[1]Department of Genetic Disease Research, Osaka Metropolitan University Graduate School of Medicine, Abeno-ku, Osaka 545-8585, Japan. [2]Guangxi Key Laboratory of Brain and Cognitive Neuroscience, Guilin Medical University, Guilin, Guangxi 541199, China. [3]Department of Neurology, Kyoto University Graduate School of Medicine, Sakyo-ku, Kyoto 606-8507, Japan. [4]Department of Neurophysiology & Neural Repair, Gunma University Graduate School of Medicine, Maebashi, Gunma 371-8511, Japan. [5]Division of Molecular Materials Science, Osaka Metropolitan University Graduate School of Science, Sumiyoshi-ku, Osaka 558-8585, Japan. [6]Department of Science and Technology, Nara Institute of Science Technology, Ikoma, Nara 630-0192, Japan. [7]Laboratory of Protein Profiling and Functional Proteomics, Osaka University Institute for Protein Research, Suita, Osaka 565-0871, Japan. [8]These authors contributed equally: Mingyue Jin, Sakiko Matsumoto, Takashi Ayaki. ✉e-mail: shinjih@omu.ac.jp

As a factor promoting this αSyn pathology, posttranslational modifications (PTMs) of αSyn have been implicated, such as phosphorylation, ubiquitination, SUMOylation, or truncation, which alter characteristics of αSyn and affect the conformation of aggregates[18–21]. The most studied PTM is phosphorylation at Ser129 (pS129) that is found on more than 90% of the αSyn in Lewy bodies from PD patients and considered an important marker for synucleinopathy[18,22]. Although the specific function of pS129 remains unclear, it seems to promote formation of αSyn filaments as well as oligomers[23,24]. However, since the multiple PTMs are mediated by the ubiquitous enzymes localized throughout the entire brain, they do not completely explain the tropism of PD pathology.

The susceptible target in PD is a catecholamine-containing neuron in the SN, Locus coeruleus or sympathetic ganglia, where tyrosine hydroxylase (TH) is specifically expressed. TH is a rate-limiting enzyme of catecholamine biosynthesis and hydroxylates the side chain of tyrosine to form dihydroxyphenylalanine (DOPA)[25]. The functional interaction between αSyn and TH has been reported, in which αSyn regulates the dopamine biosynthesis by reducing TH phosphorylation and its enzymatic activity[26–28]. However, the relationship of TH activity to αSyn aggregation kinetics or αSyn pathology in PD is still unexplored. The overlap of PD pathology with TH distribution led us to consider the possibility of TH functioning as an αSyn PTM enzyme. In this study, we show that TH hydroxylates αSyn at Tyr136 in vitro and in vivo, which facilitates oligomer formation and induces neurotoxicity, suggesting that TH contributes to PD pathogenesis.

## Results

### Identification of TH-mediated αSyn hydroxylation

To examine whether TH catalyzes the hydroxylation of tyrosine residues in αSyn, we first performed an in vitro reaction assay

(Supplementary Fig. 1a) using recombinant TH (enzymatic activity: 624 units/mg)[29] with wild-type αSyn or two mutant forms of familial PD, E46K and A53T, characterized by promoting fibril formation[30,31]. After the reaction, αSyn was purified by reverse-phase high-performance liquid chromatography (RP-HPLC). The separation profiles of TH-treated αSyn differed from those of untreated αSyn (control αSyn) in the appearance of an additional peak (Supplementary Fig. 1b). We collected αSyn fractions, including the additional peak, and digested them to make small fragments for mass spectrometry (MS). Tryptic digestion provided two tyrosine-containing fragments, namely, residues 35–43 (Tyr39) and 103–140 (Tyr125, 133, and 136) of αSyn (Fig. 1a), which were separated by RP-HPLC. The resulting separation profiles displayed two corresponding peaks in control αSyn, while TH-treated αSyn introduced several new peaks (Supplementary Fig. 2a). The larger fragment, residues 103–140, was subsequently digested by Asp-N, which provided three tyrosine-containing fragments: residues 119–125 (Tyr125), 126–134 (Tyr133), and 135–140 (Tyr136) (Fig. 1a). These fragments were also separated by RP-HPLC (Supplementary Fig. 2b). TH-treated αSyn also displayed different separation profiles accompanied by additional new peaks, implying the occurrence of some modification by TH treatment (Supplementary Fig. 2b).

We then applied the RP-HPLC eluates to MALDI-TOF MS analysis. The N-terminal peptide containing Tyr39 did not display a change in the MS signal ($m/z$ 951.4) by TH treatment (Supplementary Figs. 2c and 3a). Similarly, we could not detect the specific modifications at Tyr125 and Tyr133 in each fragment (MS peaks at $m/z$ 823.4 and $m/z$ 1069.5, respectively) (Supplementary Figs. 2c and 3b, c). On the other hand, the mass spectrum of residues 135–140 showed $m/z$ 723.2 with an additional new peak at $m/z$ 739.2 by TH treatment. This new peak exhibited an increment of 16 mass units, suggesting that potential oxidative modification occurred at Tyr136 (Fig. 1b and Supplementary

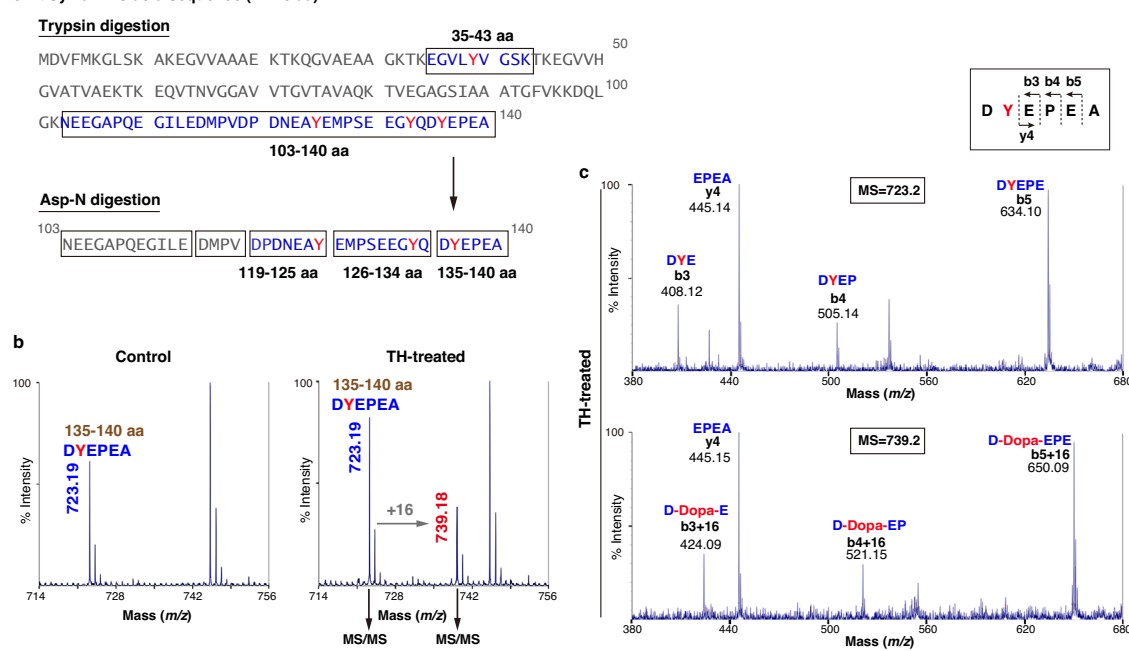

**Fig. 1 | Hydroxylation of αSyn at the Tyr136 residue by TH treatment.**
**a** Digestion process of αSyn by trypsin (upper) and Asp-N (lower). Trypsin digestion yielded two tyrosine (red)-containing fragments (blue), whose sequences were boxed. A fragment of residues 103–140 of αSyn (103–140 aa) was further digested by Asp-N, yielding five fragments, as shown by the boxed sequences. Here, Asp-N atypically cleaves the N-terminus of Glu126 but fails at N-terminus of Asp121.
**b** MALDI-TOF mass spectrum of the fragment corresponding to 135–140 aa. The peak at $m/z$ 723.19 corresponds to 135–140 aa (control, left). After TH treatment

(right), an additional peak at $m/z$ 739.18 showed an increment of 16 mass units compared to that of the 135–140 aa fragment. The peak at $m/z$ 745.2 corresponds to sodium adduct form of the 135–140 aa. **c** MALDI-TOF MS/MS spectrum of the MS peak at $m/z$ 723.2 (upper) or $m/z$ 739.2 (lower) in the TH-treated sample derived from (**b**). Fragmentation sites are indicated in the boxed area. The peptide sequences of fragmentated ions are shown above each peak, identifying the site of dopanization as Tyr136.

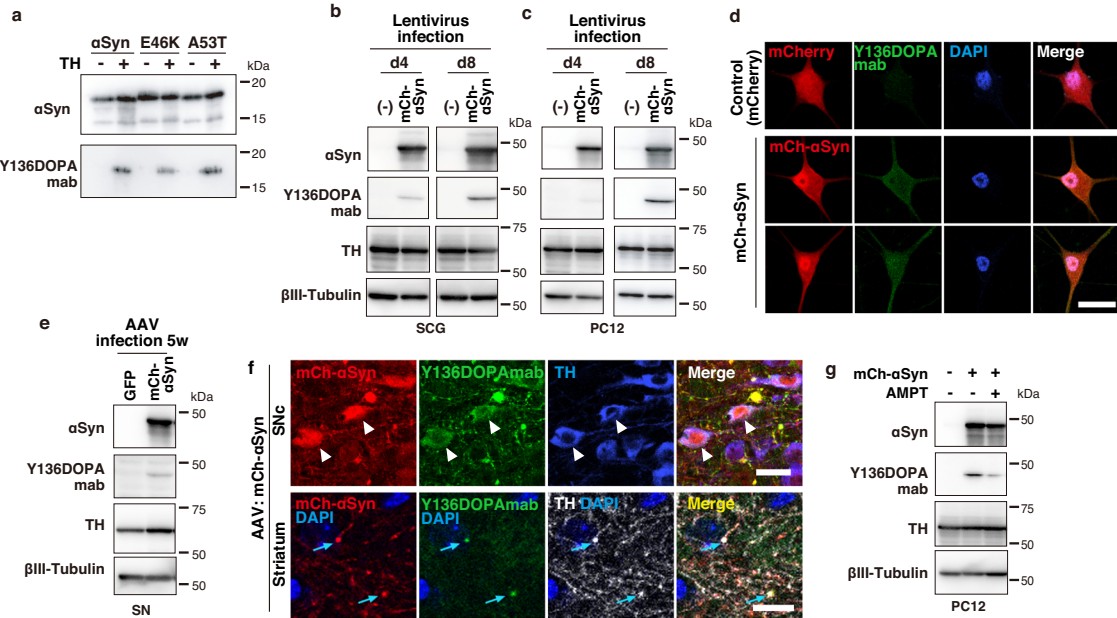

**Fig. 2 | Y136DOPA modification of overexpressed mCh-αSyn detected by a specific antibody. a** A specific antibody for Y136DOPA of αSyn. Y136DOPAmab specifically recognized TH-treated (TH +) but not control (TH-) αSyn, E46K, and A53T in WB analysis. **b, c** Y136DOPA signals detected in cultured SCG neurons (**b**) and PC12 cells (**c**). SCG and PC12 cells overexpressing mCh-αSyn by the lentivirus system were harvested at Day (d) 4 or 8 after infection and continuously examined by WB. The Y136DOPA signal was slightly detected at d4 but significantly increased at d8. βIII-tubulin is a loading control. **d** Immunofluorescence of PC12 cells overexpressing mCherry (control, upper) or mCh-αSyn (lower) by the lentivirus system. The Y136DOPA signal (green) was detected in cells expressing mCh-αSyn (red) at d8 after infection. Scale bar: 20 μm. **e** Dopanization of mCh-αSyn in mouse SN

neurons. mCh-αSyn was strongly overexpressed in the SNc by the stereotaxic injection of AAV. WB using the SN tissue detected Y136DOPA at 5 weeks after injection. **f** Immunohistochemistry staining of AAV-injected brain sections. Y136DOPA modification (green) of mCh-αSyn (red) was detected in the cell bodies of TH-positive dopaminergic neurons (blue) in the SNc (upper, white arrowheads), as well as their nerve terminals (white) in the striatum (lower, blue arrows). Scale bars: 20 μm (upper) and 10 μm (lower). **g** Y136DOPA signals decreased after treatment with the TH inhibitor. PC12 cells expressing mCh-αSyn at d10 after lentiviral infection were incubated with AMPT for 9 h and harvested. WB showed that the Y136DOPA signals were reduced after inhibiting TH enzyme activity. Data are representative of two independent experiments with similar results (**a**–**g**).

Figs. 2c and 4a, b). Amino acid sequencing by MALDI-TOF MS/MS analysis at $m/z$ 739.2 showed that Tyr136 gained 16 mass units, indicating a conversion of Tyr136 to DOPA (Y136DOPA) (Fig. 1c and Supplementary Figs. 2c and 4c, d). Concurrently, we confirmed that there was no oxidation at the C-terminal methionine Met127, which was susceptible to reactive oxygen species (ROS) produced through an oxidative reaction by TH (Supplementary Fig. 3c). These data indicate that TH catalyzes the dopanization of the intraprotein tyrosine residue Tyr136 in αSyn.

## Hydroxylation of overexpressed αSyn by endogenous TH in vivo

To obtain insights into the function of dopanized αSyn, we generated a mouse monoclonal antibody against Y136DOPA (Y136DOPAmab; Supplementary Fig. 5a) and confirmed its specificity (Fig. 2a and Supplementary Fig. 5b). Using this antibody, we first checked the modification rates at which TH converted approximately 40% of αSyn to Y136DOPA in the in vitro reaction (Supplementary Fig. 5c, d). To explore the dopanization of αSyn in cultured cells, we introduced *mCherry (mCh)-αSyn* using lentivirus-mediated gene transfer into primary sympathetic neurons isolated from the superior cervical ganglion (SCG) of mice in which endogenous TH was highly expressed. Western blot (WB) analysis using the Y136DOPAmab indicated a weak signal at Day 4 after viral infection, which was strengthened at Day 8 (Fig. 2b). Similar results were obtained for TH-positive neuronal PC12 cells overexpressing mCh-αSyn (Fig. 2c and Supplementary Fig. 6a). Immunofluorescence also indicated the Y136DOPA modification of exogenous mCh-αSyn in PC12 cells (Fig. 2d and Supplementary Fig. 6b). Furthermore, we examined the dopanization of mCh-αSyn that was overexpressed in mouse SN neurons by stereotaxic injection of adeno-associated virus (AAV; Supplementary Fig. 6c). Y136DOPA

was also detected at 5 weeks after injection by WB as well as immunohistochemistry, which labeled both the cell bodies of SN neurons and their nerve terminals in the striatum (Fig. 2e, f and Supplementary Fig. 6d–f). To examine whether this in vivo dopanization was catalyzed by endogenous TH, PC12 cells expressing mCh-αSyn were incubated with a TH inhibitor, α-methyl-*p*-tyrosine (AMPT), and analyzed by WB. After 9 hours (h) of treatment, the Y136DOPA signal was markedly decreased compared to that of inhibitor-untreated cells (Fig. 2g and Supplementary Fig. 6g), indicating that TH hydroxylation activity is required for Y136DOPA modification in vivo. In addition, we also investigated the Ser129 phosphorylation (pS129) of αSyn, a major PTM associated with αSyn pathology. Immunocytochemistry showed that the pS129 modification of mCh-αSyn overlapped with the Y136DOPA signals in PC12 cells (Supplementary Fig. 6h), suggesting that dopanization is involved in the onset of αSyn pathology. Collectively, overexpressed αSyn was posttranslationally dopanized at Tyr136 by endogenous TH in vivo.

## Dopanization of αSyn in human synucleinopathy

Next, we examined the Y136DOPA modification in a mouse model of PD. We used heterozygous A53T BAC transgenic (Tg) mice that express human A53T similarly to the endogenous αSyn expression pattern and display the degeneration of SN neurons in an age-dependent manner[32]. Immunohistochemistry staining using Y136DOPAmab did not indicate any signals in the cell bodies at SN pars compacta (SNc) from either 18-month-old wild-type or Tg mice (Fig. 3a). In contrast, the striatum to which SN neurons were projecting showed punctate signals in Tg mice but not in wild-type mice (Fig. 3a). Furthermore, a similar analysis was performed on brain sections from patients with α-synucleinopathies, PD and multiple system atrophy (MSA), revealing a significant increase

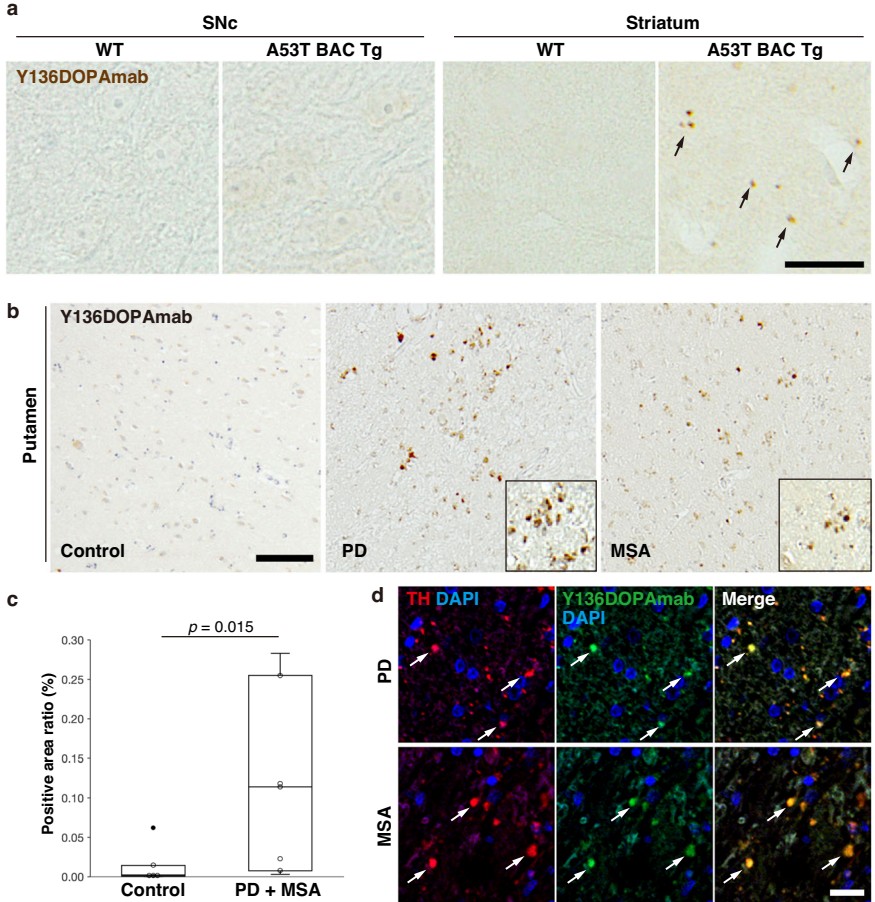

**Fig. 3 | Y136DOPA modification in A53T BAC Tg mice and human α-synucleinopathies. a** Y136DOPA signals (brown) in A53T BAC Tg mice. Eighteen-month-old wild-type (WT) and Tg mice were analyzed by immunohistochemistry staining using Y136DOPAmab. Tg mice, but not WT mice, showed punctate signals in the striatum (black arrows), whereas the cell bodies in the SNc did not show any signals in either mouse genotype (left panels). Scale bar: 20 μm.
**b**, **c** Immunohistochemical analysis of the putamen from human α-synucleinopathies. The sections from patients with PD (middle) and MSA (right) and age-matched control (left) were immunostained and displayed an increase of Y136DOPA signals (brown) in patient sections compared with that of control (**b**). Magnified images are shown in the corners of the PD and MSA panels. Nuclei were counterstained with hematoxylin (purple). The positive area ratio of these signals was significantly increased in the PD + MSA group (**c**). Box-and-whisker plots definitions: center line, median; box limits, upper and lower quartiles; whiskers, 1.5x interquartile range; black point, outlier. Sample size: $n = 7$ per group (PD $n = 3$, MSA $n = 4$). $P$-value was calculated using two-sided Wilcoxon test. Scale bar: 100 μm.
**d** Immunofluorescence of Y136DOPA. Putamen sections of PD (upper) and MSA (lower) brains were double-immunolabeled with Y136DOPAmab (green) and anti-TH antibody (red). Y136DOPA-positive signals were localized in the nerve terminals of dopaminergic neurons projected from the SNc (white arrows). Scale bar: 20 μm. Data are representative of two independent experiments with similar results (**a**, **d**).

in the Y136DOPA-positive area in the posterior putamen of patients compared with that of the control sections (Fig. 3b, c). These signals were colocalized with the TH-positive axon terminals of the SN neurons in the putamen (Fig. 3d), indicating that dopanized αSyn was distributed in the presynapses of dopaminergic neurons in PD and MSA cases. These results suggest that TH-mediated Y136DOPA modification is involved in the pathogenesis of human α-synucleinopathies.

## Oligomer assembly of αSyn triggered by Y136DOPA modification

An increasing amount of evidence suggests that the oligomeric and fibrillar aggregation of αSyn plays a central role in the pathogenesis of PD and other synucleinopathies[33]. In particular, the oligomeric form of αSyn has a higher cytotoxicity, leading to the disruption of biological membrane integrity[15]. To examine the effects of Y136DOPA modification on fibrillization, recombinant FLAG-tagged αSyn (f-αSyn, f-E46K, or f-A53T) with or without prior TH-mediated dopanization was incubated with continuous agitation for 18 h. After incubation, we confirmed that both control and dopanized f-αSyn were detected by an antibody, Syn-O2, that recognized both oligomeric and fibrillar conformations of αSyn[34] in a dot blot assay (Supplementary Fig. 7a). A solubility assay using these oligomeric/fibrillar f-αSyn showed that sarkosyl-insoluble f-αSyn aggregates were reduced by TH treatment (Supplementary Fig. 7b). We then examined the structures of f-αSyn by transmission electron microscopy (TEM) after 18 h of incubation and found that control f-αSyn formed large fibril aggregates. On the other hand, to our surprise, dopanized f-αSyn formed short, dispersed oligomer-like structures (Fig. 4a). Similarly, dopanized f-E46K and f-A53T also formed separated short oligomer-like assemblies (Supplementary Fig. 7c). These data suggest that the dopanization of Tyr136 prevents the assembly of large aggregates and stabilizes small oligomer-like conformations. We further used sucrose gradient centrifugation to estimate the molecular weight of the f-αSyn assemblies. Before 18 h of incubation, both control and dopanized f-αSyn displayed a small molecular mass regardless of TH treatment (Fig. 4b and Supplementary Fig. 7d). After 18 h of incubation, control f-αSyn formed aggregates with higher molecular masses, while dopanized f-αSyn still retained smaller masses, similar to the TEM images (Fig. 4b and Supplementary Fig. 7d). To confirm the oligomer formation of

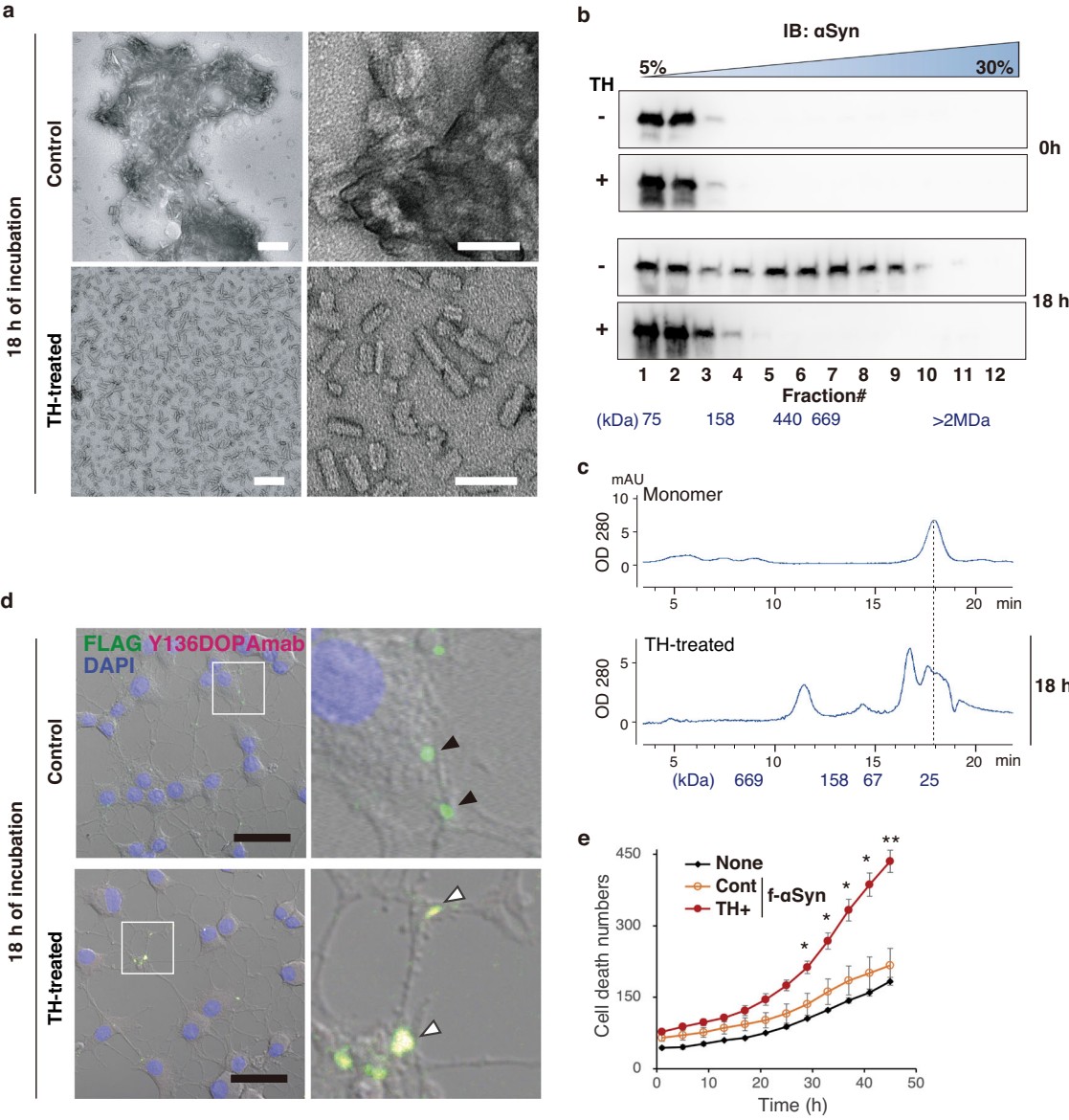

**Fig. 4 | Oligomer formation and cytotoxicity of dopanized f-αSyn. a** Negatively stained TEM images of FLAG-tagged αSyn (f-αSyn). Control (upper) and TH-treated (lower) f-αSyn were incubated for 18 h before staining with 2% uranyl acetate. Control f-αSyn formed large fibril clusters, whereas dopanized f-αSyn tended to form a separated oligomeric conformation. High-magnification images are shown on the right side of each panel. Scale bars: 200 nm (left) and 50 nm (right). **b** WB analysis of the sucrose gradient fractionation of f-αSyn. Before (0 h) or after (18 h) incubation, control (TH−) and dopanized (TH+) f-αSyn were separated by 5–30% sucrose gradient centrifugation. Twelve 1-mL fractions were examined by WB using an anti-αSyn antibody. The corresponding sucrose concentrations are indicated at the top of the panels. After 18 h of incubation, control f-αSyn formed aggregates with large masses, whereas dopanized f-αSyn retained a smaller mass. **c** SEC of dopanized f-αSyn oligomers. The chromatogram of SEC using monomeric (upper) or TH-treated f-αSyn after 18 h of incubation (lower) showed that dopanized f-αSyn

formed oligomers up to 15-mers. **d** Detection of f-αSyn uptake in PC12 cells. Control (upper) and TH-treated (lower) f-αSyn after 18 h of incubation were added to PC12 cells for 24 h and immunostained with anti-FLAG antibody (green) and Y136DOPAmab (magenta). Rectangle-surrounding areas were enlarged on the right side. Black arrowheads indicate control f-αSyn taken up into cells, and white arrowheads point to dopanized f-αSyn (yellow). Scale bar: 20 μm. **e** Real-time quantification of cell death in PC12 cells. The numbers of dead cells were counted using IncuCyte Cytotox Green Dye, resulting in a significant increase in cytotoxicity by treatment with dopanized f-αSyn (TH+) compared with control f-αSyn (Cont). Data are presented as mean ± SEM. Sample size: $n = 3$ well/treatment. $P$-values were calculated using two-tailed unpaired $t$-test between Cont and TH+ groups (29 h *$p = 0.044$; 33 h *$p = 0.027$; 37 h *$p = 0.018$; 41 h *$p = 0.011$; 45 h **$p = 0.007$). None indicates the sample without any αSyn treatment. Data are representative of two independent experiments with similar results (**a, b, d**).

small-mass dopanized f-αSyn, we performed size exclusion chromatography (SEC) and compared the results with those of monomeric recombinant αSyn. The resulting chromatograms showed that dopanized f-αSyn formed oligomers up to 15-mers after 18 h of incubation (Fig. 4c and Supplementary Fig. 7e). These data support that Y136DOPA modification of αSyn promotes the maintenance of a small oligomeric state and prevents the formation of large-mass aggregation.

## Cytotoxicity of dopanized αSyn

Cell-to-cell propagation of αSyn aggregates in a prion-like manner is thought to contribute to the progression and spreading of αSyn neuropathology in synucleinopathy[35]. We investigated the cellular uptake and toxicity of dopanized f-αSyn after 18 h of incubation using PC12 cells. At 24 h after the addition of f-αSyn to the culture medium, dopanized f-αSyn was internalized into cells as well as control f-αSyn (Fig. 4d and Supplementary Fig. 8a). Then, we examined the

cytotoxicity of f-αSyn by the real-time quantification of cell death using the IncuCyte Live-Cell Analysis System. PC12 cells treated with dopanized f-αSyn exhibited a significant increase in cell death in a time-dependent manner compared to that of cells treated with control f-αSyn (Fig. 4e and Supplementary Fig. 8b). Collectively, these results demonstrate that dopanized αSyn has a higher neurotoxicity than control αSyn, suggesting that dopanized αSyn and its oligomer species may contribute to the selective loss of dopaminergic neurons in the PD brain.

## Discussion

In this study, we discovered a PTM of αSyn in which TH converts Tyr136 to DOPA in SN neurons. Although TH has been broadly recognized as an enzyme for the hydroxylation of free tyrosine molecules[36], our results indicate the possibility that the intraprotein tyrosine residue is also an enzymatic substrate for TH. Catalytic domains of enzymes are mainly buried inside the protein tertiary structure to ensure their substrate specificities[37]. However, the catalytic iron atom of TH is located near the surface, 10 Å below the enzyme surface within a 17 Å-deep active-site cleft[38] (Supplementary Fig. 9a). Grid-based HECOMi finder (GHECOM) calculations also show the presence of a large groove containing the catalytic site on the TH surface (Supplementary Fig. 9a). On the other hand, the crystal structure of αSyn shows that Tyr136 is located in its flexible C-terminal tail (Supplementary Fig. 9b). This intrinsically disordered (ID) C-terminal region does not form a three-dimensional structure under physiological conditions and even in fibril constructs, likely allowing Tyr136 to fit into the TH catalytic site[39]. In contrast, Tyr39 located between the two α-helices probably does not become a substrate for TH due to the structural restriction. ID domains in proteins are generally susceptible to PTM[40,41], which changes their physicochemical properties and induces the alteration of their conformation. Similarly, Y136DOPA modification caused significant changes in αSyn aggregation kinetics. Based on previous works and our findings, we speculate that other proteins with flexible ID domains may be also subjected to TH-mediated hydroxylation for modifying their biochemical properties. Further study is needed to confirm the generality of dopanization as a PTM.

In the in vitro reaction assay and MS analysis, we were unable to detect dopanization at two tyrosine residues in the C-terminal region, Tyr125 and Tyr133, despite their proximity to Tyr136. This result may imply the presence of a preferential sequence for dopanization. Another possibility is that the lower stability of DOPA-containing fragments under the experimental conditions might make MS detection difficult. Catechol compounds are prone to releasing ROS in the presence of transition metals and oxygen, which induces nonenzymatic peptide degradation[42,43]. Indeed, the RP-HPLC peak corresponding to C-terminal region of αSyn showed a remarkable reduction in height after TH treatment (Supplementary Fig. 2a). Considering the influence of ROS from the "DOPA residue" on peptide stabilities, the possibility of dopanization at Tyr125 and Tyr133 is not completely precluded.

Amyloid aggregation of αSyn in the SN is closely associated with the selective loss of dopaminergic neurons, resulting in defective physical movements[44,45]. However, recent findings have demonstrated that oligomeric αSyn species exacerbate αSyn-mediated toxicity[46,47]. Moreover, the injection of sonicated short αSyn fibrils into mouse striatum propagates Lewy pathology and results in the progressive loss of dopaminergic neurons[35]. The Y136DOPA modification of αSyn induced the formation of an oligomeric structure and displayed a higher cytotoxicity to PC12 cells than the control. Some PTMs, such as C-terminal truncation and phosphorylation, have been implicated in αSyn oligomer formation and neurotoxicity[18,21]. The negatively charged C-terminus and phosphorylation at Ser129 may act to retard fibril formation with a β-sheet amyloid structure[48,49]. Presumably, the

dopanization of αSyn at Tyr136 augments its hydrophilicity and thereby triggers the formation of oligomers, which may play a pathogenic role in dopaminergic neurodegeneration. The Y136DOPA signal was also detected at the striatonigral axon terminals of aged A53T Tg mice and patients with PD and MSA. These results strongly suggest the involvement of αSyn dopanization in pathology in vivo. On the other hand, neither control mice nor control human brains showed any dopanized αSyn signal, which may suggest the possibility that the dopanized αSyn is physiologically undetectable and accumulates depending on pathological progression. Altogether, our findings implicate that TH participates in αSyn pathology, providing a potential explanation for the selective loss of dopaminergic neurons in synucleinopathies.

The familial PD-linked pathogenic mutations act to accelerate disease onset and progression[4,5,50]. In the fibrillization assay, we found no obvious differences in the effect of dopanization among wild-type and mutated αSyn in this study. Previous works have revealed that E46K and A53T accelerate the formation of large fibrillar aggregates[30,31]. Consistently, control E46K and A53T without TH treatment formed aggregates with larger masses compared to that of the wild-type, whereas the dopanization made wild-type αSyn and two mutants assemble into the short oligomeric forms (Fig. 4b and Supplementary Fig. 7d). Presumably, the increased hydrophilicity caused by dopanization may contribute to abolishing this aggregation preference in mutated αSyn, suggesting that dopanization is a critical factor in promoting oligomer formation. In addition to point mutations, the increases in αSyn protein levels and aging has also been considered a major factor in PD, which promotes the abnormal accumulation of misfolded and stabilized αSyn[51–53]. Altogether, the aberrant metabolism of αSyn forms the foundation of PD, and the dopanization of αSyn may trigger the preferential degeneration of dopaminergic neurons. Further analysis may help us to precisely explain how this new PTM contributes to PD pathogenesis. Chemical compounds that protect the dopanization of αSyn but do not suppress free DOPA synthesis by TH may be applicable in preventing the progression of PD. Our work will open the way to drug discovery for the therapeutic intervention of PD.

## Methods

All methods were performed in accordance with the approved guidelines. All animal studies were approved by the Institutional Animal Care and Use Committee, Osaka Metropolitan University (#21094), and Kyoto University (#Med Kyo 21001). Recombinant DNA experiments were approved by the Safety Committee for Experiments using Living Modified Organism, Osaka Metropolitan University (#708). Experiments with human samples were approved by the Ethics Committee of Kyoto University (#R1038).

### Plasmids

For the expression of recombinant proteins in *E. coli*, cDNA of human TH, human αSyn, E46K and A53T, and FLAG-tagged human αSyn and two mutants (f-αSyn, f-E46K, f-A53T) was subcloned into the pGEX-6P expression vector (Cytiva). Mutations of αSyn were introduced by the PrimeSTAR Mutagenesis Kit (Takara) or direct synthesis of entire cDNA (Invitrogen). FLAG tag was fused at N-terminus of αSyn or mutants in frame. For lentivirus-mediated expression, three plasmids, pCAG-HIVgp, pCMV-VSV-G-RSV-Rev, and pCSII-EF-MCS-IRES2-Venus, were prepared as previously described[11]. pCSII-EF-MCS-IRES2-Venus was modified by replacing the IRES2-Venus cassette with mCherry or mCherry (mCh)-tagged human αSyn (mCh-αSyn, mCh-E46K, mCh-A53T). mCh-αSyn were constructed by inserting αSyn at the C-terminus of mCherry using the pmCherry-C1 vector (Takara). To produce an adeno-associated virus (AAV) serotype 9 vector that expressed mCh-αSyn, mCh-E46K, or mCh-A53T, we replaced GFP with mCh-αSyn or its mutants in pAAV-SynImCMV-GFP-WPRE-SV40polyA,

which carries GFP, woodchuck hepatitis posttranscriptional regulatory element (WPRE) and simian virus 40 polyadenylation signal (SV40polyA) downstream of the synapsin I promoter with a minimal CMV promoter sequence (SynImCMV)[54].

## Recombinant proteins

GST-conjugated αSyn, E46K, A53T, f-αSyn, f-E46K, f-A53T, and TH were expressed in the *E. coli* Rosetta strain (Novagen) cultured in L-Broth supplemented with 0.1 mM IPTG at 20 °C for 12 h. GST proteins were extracted from bacteria by sonication and purified using GST-Sepharose (Cytiva) according to standard procedures under the condition of 20 mM HEPES, pH 7.4, 150 mM KCl, 0.1% NP-40, and 0.5 mM DTT. To remove the GST tag, GST-TH and some GST-αSyn were treated with PreScission protease (Cytiva). Purified TH, αSyn, GST-αSyn, and GST-f-αSyn proteins were frozen and cryopreserved at −80 °C until use.

TH enzymatic activity was measured using QDPR (quinoid dihydropteridine reductase) and NADH (nicotinamide adenine dinucleotide reduced form) as described previously[55] with some modifications. Briefly, TH (10 μM) and tyrosine (0.3 mM) interaction was conducted in 20 mM MES, pH 6.2, 0.1 mM tetrahydro-L-biopterin (BH₄; Wako), 25 μM FeSO₄, 0.5 μM QDPR, and 0.15 mM NADH in a 3 mL reaction volume. Incubation was performed for 10 min at 37 °C. In this reaction, TH converted tyrosine and $BH_4$ to DOPA and dihydro-L-biopterin ($BH_2$), respectively. $BH_2$ was recycled to $BH_4$ by QDPR through NADH-mediated reduction[56]. NAD yield was measured by absorbance at 340 nm, and TH enzymatic activity was calculated.

## In vitro reaction of αSyn and TH

The in vitro reaction of TH (10 μM) and αSyn or two mutants (5 μM each) was conducted in 20 mM MES, pH 6.2, 100 μM $BH_4$, and 25 μM $FeSO_4$ in a 50 μL reaction volume. Incubation was performed for 10 min at 37 °C. The negative control described as 'Control' in this study contained neither TH nor $BH_4$. For RP-HPLC and fibrillization, GST-αSyn and GST-f-αSyn were used, respectively, and separated from the reaction mixture using GST-Sepharose. Binding to beads in 150 mM KCl and cleavage by PreScission protease were performed as described above.

## RP-HPLC and MS analysis

The above αSyn and its mutants were purified by using RP-HPLC (L-7100, Hitachi LaChrom HPLC System). The proteins were loaded on a column (Develosil C-4 HG column, Nomura Chemical) that was equilibrated with 95% solvent A (0.1% trifluoroacetic acid (TFA)-H₂O)/5% solvent B (0.1% TFA-acetonitrile (ACN)) and eluted using a linear gradient of solvent B (5–80%) in 40 min at a flow rate of 1 mL/min. Detection was performed by UV absorption at 220 nm. The eluted αSyn fraction was dried by a vacuum centrifugal concentrator, dissolved in 100 mM ammonium carbonate and 10% ACN, and digested with 1 pmol/μL trypsin (Promega). The digested products were subjected to a second RP-HPLC (HP1100 HPLC system, Agilent Technologies) equipped with a ZORBAX 300SB-C8 (Agilent Technologies), and eluted using the same elution gradient as the first RP-HPLC in 20 min at a flow rate of 0.32 mL/min. Detection was performed by UV absorption at 280 nm. The fraction containing the C-terminal fragment (residues 103–140) of αSyn was dried and digested by 4 ng/μL Asp-N (Roche), followed by a third RP-HPLC separation using the same chromatographic conditions and column as the second RP-HPLC separation. Eluted 0.5-min fractions were dried and dissolved in 10% ACN for MS analysis.

Each fraction was spotted onto a stainless steel plate, dried, and mixed with a matrix solution (0.5 μL per spot; 7 mg/mL of α-cyano-4-hydroxycinnamic acid (Sigma) in 50% ACN, 0.1% TFA). MALDI-TOFMS and MS/MS measurements were carried out with a 4700 MALDI-TOF/TOF mass spectrometer (Applied Biosystems) as previously described[57]. All mass spectra were obtained by averaging 2500 laser shots from each sample well in positive-ion mode. The instrument was calibrated with a mixture of peptides, angiotensin I (*m/z* 1296.6), dynorphin (*m/z* 1604.0), ACTH (1–24) (*m/z* 2932.6), and β-endorphin (*m/z* 3463.8). For the MS/MS measurements, the metastable suppressor and CID gas were both set to "ON". The entire process was controlled using 4000 series Explorer software (version 3.6; Applied Biosystems). Spectra were processed and analyzed using Data Explorer software (version 4.8; Applied Biosystems).

## Generation of mouse monoclonal antibody

Peptide corresponding to the C-terminal residues 131–140 of human αSyn with additional cysteine at its N-terminus (Cys-EGYQDYEPEA, 136Y peptide), antigen peptide with DOPA instead of Tyr136 (Cys-EGYQD-Dopa-EPEA, 136DOPA peptide) and keyhole limpet hemocyanin (KLH)-conjugated 136DOPA peptide (KLH-136DOPA) were synthesized by Peptide Institute, Inc. Immunogen was prepared by mixing 2 mg/mL KLH-136DOPA with an equal volume of TiterMax Gold (TiterMax) to form a stable emulsion. We immunized twenty 7-week-old female BALB/c mice with the injection of immunogen emulsion subcutaneously into both hind footpads and back skin (total 100 μg/mouse). Mice were maintained at 25 °C with 55% humidity on a 12-h light-dark cycle, given free access to food and drinking water, and handled according to the institutional guideline for animal experiments. After 2 weeks, mice were subjected to the second immunization with an emulsion of KLH-136DOPA (50 μg/mouse) with Freund's incomplete adjuvant (Wako). After 1 week, a small amount of serum was taken from each immunized mouse to test the immune response against the 136DOPA peptide by ELISA. At 3 weeks after the second immunization, animals were administered 1 mg/mL KLH-136DOPA in PBS, pH 7.4, intraperitoneally as the final boost (150 μg/mouse), followed by hybridoma generation at 4 days after booster injection.

Prior to hybridoma generation, mouse myeloma cells (Sp2/O-Ag14, JCRB9084, JCRB Cell Bank) were maintained in a 75 cm² flask with GIT medium, 10% FBS, and P/S. The spleen of immunized mice was dissected and dissociated from the capsule. Splenocytes and mouse myeloma cells were fused at a ratio of 5:1 by the addition of 1 mL of PEG 1500 (Roche) gently over 1 min with agitation. After incubation for 1 min, 15 mL of DMEM was slowly added over 5 min. Then, cells were collected and resuspended in HAT conditioned medium (GIT, 100 μM hypoxanthine, 0.4 μM aminopterin, 16 μM thymidine, 5% BM Condimed H1 (Roche) and P/S), spread in 96-well plates (2 × 10⁵ splenocytes/well) and cultured for 9 days in a CO₂ incubator at 37 °C before ELISA.

## Enzyme-linked immunosorbent assay (ELISA)

For hybridoma screening, we diluted 136Y and 136DOPA peptide to 3 μg/mL with PBS and added 50 μL/well of this solution to 96-well immunoplates (Thermo Scientific), followed by the incubation for 2 h at 37 °C. After washing with PBS, blocking buffer (0.5% BSA in PBS) was added to each well and incubated for 1 h at 37 °C. After washing plates with PBS, hybridoma culture supernatant was applied onto each plate precoated by 136Y or 136DOPA peptides and incubated overnight at 4 °C. Plates were washed with PBS 4 times and incubated with Peroxidase AffiniPure Donkey Anti-Mouse IgG (1:3000, Jackson Immunoresearch, 715-135-150) diluted with 0.1% BSA in PBS for 1 h at 37 °C. After washing, 75 μL of developing buffer (0.4 mg/mL o-enylenediamine dihydrochloride (OPD), 0.012% H₂O₂, 0.05 M phosphate-citrate buffer, pH 5.0) was added to each well. The color reaction was stopped by the addition of 25 μL of 3 M H₂SO₄.

For quantification of dopanized αSyn in the in vitro reaction, a serial dilution of 136DOPA peptide (4, 8, 16, or 32 nM in PBS) was subjected to ELISA using Y136DOPAmab. After stopping the reaction, the OD at 490 nm was measured, and a standard curve was established.

Control and TH-treated αSyn after the in vitro reaction were purified by RP-HPLC, diluted at concentrations of 24 or 48 nM, and tested by ELISA using Y136DOPAmab. The amount of dopanized αSyn at Tyr136 by TH was calculated based on the standard curve.

## Lentiviral packaging

Lentiviral vectors were produced as described previously[11]. Briefly, Lenti-X-293T cells (632180, Takara) were prepared on collagen-coated dishes with DMEM, 10% FBS, and P/S. The following day, cells were cotransfected with the three plasmids described above using poly-ethyleneimine (Sigma). After 48 h, culture medium containing the virus particles was harvested, centrifuged at $2000 \times g$ for 10 min, filtered through 0.45-μm filters (Millipore) to remove cell debris, and concentrated by ultracentrifugation (Himac, Hitachi) using a swing rotor at $52,000 \times g$ for 2 h. The virus particles were resuspended in sterile PBS and stored at −80 °C until use.

## Cell culture

For sympathetic neuron primary culture, superior cervical ganglions (SCGs) were dissected from twenty postnatal day 0–2 male or female C57BL/6 mice (ten mice/experiment) and transferred to L15 medium. After centrifugation, SCGs were incubated with 1 mg/mL collagenase at 37 °C for 1 h, followed by treatment with 0.25% trypsin and 0.1% DNase at 37 °C for 20 min. After trituration, cells were spread on collagen-coated 12-well plate (cells from 5 ganglions/well) with DMEM/Ham's F12 medium supplemented with N-2 supplement (Invitrogen), 100 ng/mL murine NGF (Promega), GlutaMAX (Invitrogen) and P/S. Half the volume of culture medium was exchanged with fresh medium every 2 days. At 2 days in vitro (DIV), the lentiviruses carrying mCh-αSyn were added to the SCG culture for over-expression. To prevent glial proliferation, cytosine-1-β-D-arabino-furanoside (AraC, Sigma) was added at a 0.25 μM concentration to cultures at DIV 4. SCGs were harvested sequentially at DIV 6 or 10 (Days 4 or 8 after viral infection, respectively) and subjected to WB analysis.

For PC12 cell culture, cells were obtained from JCRB Cell Bank (JCRB0733) and cultured on PLL and laminin-coated round-glass coverslips ($\varphi = 18$ mm) in 12-well plates ($4 \times 10^4$ cells/well) and collagen-coated 6-well plates ($5 \times 10^4$ cells/well) for immunocytochemistry and WB, respectively, with RPMI 1640 containing 10% FBS, 5% heat-inactivated horse serum (HS), GlutaMAX and P/S. At DIV 1, the medium was exchanged with the differentiation medium (RPMI 1640 containing 0.5% FBS, 0.25% HS, 50 ng/mL NGF, GlutaMAX, and P/S). Half the volume of culture medium was exchanged with fresh medium every 3 days. At DIV 4, cells were infected by lentivirus carrying mCherry, mCh-αSyn, mCh-E46K, or mCh-A53T. At DIV 8, 12, or 15 (Days 4, 8, or 11 after viral infection, respectively), PC12 cells were harvested for WB analysis. Cells at DIV 12 (Day 8 after viral infection) were also fixed for immunocytochemistry. For inhibition of TH activity, cells were treated by 1 mM α-methyl-p-tyrosine (Sigma) for 9 h at DIV 14 (Days 10 after viral infection) and harvested for WB analysis.

## Immunocytochemistry

The above PC12 cells were fixed with 4% paraformaldehyde (PFA), rinsed with PBS, and permeabilized with 0.1% Triton X-100 in PBS. After blocking with Block-Ace (KAC) and 1% BSA in PBS, cells were incubated with mouse Y136DOPAmab and rabbit anti-αSyn pS129 (1:1000, Abcam, ab168381) overnight at 4 °C. After washing with PBS, cells were incubated with Goat anti-Mouse IgG Secondary Antibody, Alexa Fluor 488 (1:1000, Invitrogen, A11001), Goat anti-Rabbit IgG Secondary Antibody, Alexa Fluor 647 (1:1000, Invitrogen, A21244) and DAPI (1:1000, Dojindo) in blocking buffer for 1 h at room temperature (RT). After rinsing with PBS, cells on coverslips were mounted. Fluorescent images were acquired using an LSM700 confocal microscope (Carl Zeiss) and ZEISS ZEN software.

## AAV-stereotaxic injection and immunohistochemistry

AAV serotype 9 vectors for the expression of GFP, mCh-αSyn, or its mutants were produced as described previously[58]. Anesthetized six-teen 9-week-old male or female C57BL/6 mice (four mice/AAV) were placed in a stereotaxic frame. AAV solutions at a volume of 0.7 μL were injected through a glass capillary, unilaterally targeting the left SN at the following coordinates: 1.3 mm rostral relative to lambda, 1.5 mm lateral, and 4.3 mm ventral. The glass capillary was left in place for 5 min before it was slowly retracted.

Three weeks after surgery, mice were transcardially perfused with Zamboni solution (2% PFA, 0.2% picric acid in 0.1 M PB, pH7.4). Brains were removed, postfixed in Zamboni solution overnight, and placed in 30% (w/v) sucrose in 0.1 M PB for 2 days, followed by freezing in powdered dry ice. Coronal sections of the SN and striatum were cut using a cryostat (CM1950, Leica Microsystems) at 16-μm thickness. For immunohistochemistry, sections were dried and washed with PBS. Antigen retrieval was carried out in 10 mM citric buffer, pH 6.0, at 80 °C for 20 min. After blocking with 1% BSA and 0.3% Triton X-100 in PBS for 1 h, sections were incubated with mouse Y136DOPAmab and rabbit anti-TH antibody (1:1000, Millipore, AB152) overnight at 4 °C. After washing with PBS, sections were incubated with Goat anti-Mouse IgG Secondary Antibody, Alexa Fluor 488 (1:1000, Goat anti-Rabbit IgG Secondary Antibody, Alexa Fluor 488 (1:1000, Invitrogen, A11008), Goat anti-Rabbit IgG Secondary Antibody, Alexa Fluor 647 (1:1000) and DAPI (1:3000) in blocking buffer at RT for 2 h. After rinsing with PBS, sections were mounted. Microscope images were acquired using LSM700 confocal microscopy and ZEISS ZEN software.

## Immunohistochemistry of transgenic (Tg) mice

A53T BAC Tg mice were generated previously[32]. Three male or female C57BL/6 and Tg mice (18 months old) were anesthetized with sevo-flurane and intracardially perfused with cooled PBS followed by 4% PFA. The brain samples were postfixed with 4% PFA and embedded in paraffin. Coronal sections of the SN and striatum were cut on a microtome (EG1150, Leica Microsystems) at 8-μm thickness. The sections were pretreated in 100% formic acid at RT for 15 min and then in 0.1 M PBS at 120 °C for 10 min using an autoclave. For quenching endogenous peroxidase, sections were treated with 3% (v/v) $H_2O_2$ in PBS for 30 min at RT. After blocking using mouse blocking reagent (414322, Histofine Mouse stain kit, Nichirei Biosciences) for 1 h at RT, sections were then incubated with Y136DOPAmab overnight at 4 °C, followed by appropriate polymer secondary antibody (414322, Histofine mouse stain kit, Nichirei Biosciences). Between steps, the sections were washed three times with 0.1 M PBS for 5 min each and visualized using a peroxidase stain DAB kit (25985-50, Nakarai Tesque). The images of interest were captured with a microscope (BX43, Olympus).

## Immunohistochemistry of human brain

Formalin-fixed human brain tissue samples from 3 patients with MSA (71–78 years old), 4 patients with PD (67–78 years old), and 7 age-matched controls (62–86 years old) without significant αSyn disease and clinical history of parkinsonism were used in this study, as shown in Supplementary Table 1. For the study with autopsied materials, all the bereaved family signed the written informed consent, which includes that the donated human autopsied tissues and clinical information will be used for academic conferences and scientific paper presentations. To obtain the corresponding portion from each case, the posterior putamen was obtained from coronal sections containing the globus pallidus. Patient diagnosis was determined by clinical information and pathological examination.

For DAB staining, 6-μm paraffin sections were pretreated at 120 °C for 20 min in Histofine deparaffinizing antigen retrieval buffer, pH 6 (415281, Nichirei Biosciences), using an autoclave, followed by over-night incubation with Y136DOPAmab in PBS containing 3% BSA at 4 °C. The sections were incubated with peroxidase polymer secondary

antibody (424154, Histofine Simple Stain MAX-PO MULTI, Nichirei Biosciences) and developed with a DAB Substrate Kit (SK 4100, Vector Laboratories). After immunostaining, sections were counterstained with hematoxylin and cover slipped. For quantification of Y136DOPAmab-positive area densities, pictures under 200× magnification for each slide to encompass the lateral part of the putamen from each case were taken. The positive area percentage was calculated for each picture automatically using Fiji software (https://fiji.sc/).

For immunofluorescence, sections were incubated with Y136DOPAmab and rabbit anti-TH antibody (1:500), followed by incubation with Goat anti-Mouse IgG Secondary Antibody, Alexa Fluor 488 (1:200) and Goat anti-Rabbit IgG Secondary Antibody, Alexa Fluor 546 (1:200, Invitrogen, A11010). The slides were cover slipped with Vectashield containing DAPI (Vector Laboratories) and were then viewed with a FLUOVIEW FV-1000 confocal laser scanning microscope (Olympus).

### SDS–PAGE and western blotting (WB)

Protein extraction from SCG neurons, PC12 cells and mouse SN tissues was performed using lysis buffer (50 mM Tris-HCl, pH 7.5, 150 mM NaCl, 1 mM EDTA, and 1% Triton X-100), followed by low-intensity sonication pulses. The above protein extracts or HPLC-purified αSyn after the in vitro reaction were separated by 12.5% or 15% acrylamide gels, respectively, followed by transfer to PVDF membranes or Coomassie brilliant blue staining. Blots were incubated with blocking buffer (5% skim milk in TBS-T) and immunostained overnight at 4 °C with the following primary antibodies: mouse anti-αSyn (1:1000, BD Transduction Laboratories, 610787), rabbit anti-TH (1:1000), mouse anti-RFP (1:2000, MBL, M208-3), mouse anti-βIII-tubulin (Tuj1, 1:2000, R&D Systems, MAB1195) antibodies and mouse Y136DOPAmab. After washing with TBS-T, the membranes were incubated with Peroxidase AffiniPure Donkey Anti-Mouse IgG (1:3000) or Peroxidase AffiniPure Donkey Anti-Rabbit IgG (1:3000, Jackson ImmunoResearch, 711-035-152) for 1 h at RT. After washing, the immunoblots were developed using ECL Prime detection reagent (Cytiva). Chemiluminescence was detected by Fusion solo S (VILBER). The uncropped blots images are provided in the Source Data file.

For the antibody absorption assay, Y136DOPAmab was preincubated with 1 μg/mL 136DOPA or 136Y peptides overnight at 4 °C before use.

### Fibrillization of f-αSyn and transmission electron microscopy (TEM)

f-αSyn, f-E46K, and f-A53T after the in vitro reaction were purified by RP-HPLC, vacuum-dried, and dissolved in 10 mM MES containing 150 mM KCl, pH 6.2, to make 7 μM (~0.1 μg/μL) concentration before incubating at 37 °C for 18 h with continuous agitation. For TEM, 5 μL of each sample was placed on a carbon film grid (400-mesh) and negatively stained with 2% uranyl acetate. Micrographs were collected by a Talos F200C (Thermo Fisher) at 22,000× (0.478 nm/pixel) and 57,000× (0.184 nm/pixel) magnification.

### Dot blot analysis

The control and TH-treated f-αSyn before and after fibrillization prepared above were spotted on nitrocellulose membranes (5 μL each), followed by blocking with 5% skim milk for 1 h. These membranes were subsequently incubated with mouse anti-αSyn, mouse anti-FLAG (1:500, Sigma, F1804), mouse anti-αSyn, aggregated (Syn-O2, 1:500, BioLegend, 847602) antibodies, and mouse Y136DOPAmab for 1 h at RT. After washing with TBS-T, the membranes were treated with Peroxidase AffiniPure Donkey Anti-Mouse IgG (1:3000) for 1 h, washed with TBS-T, and developed using ECL Prime detection reagent.

For the f-αSyn solubility assay, f-αSyn and its mutants after fibrillization were centrifuged at 627,000 × g (120,000 rpm, Beckman Optima MAX-TL ultracentrifuge with TLA 120.2 rotor) for 1 h. The pellet was dissolved in 100 μL of 2% sarkosyl, 10 mM MES, pH 6.2, and 150 mM KCl and centrifuged at 627,000 × g for 20 min. The pellet was dissolved in 50 μL of 10 mM MES, pH 6.2, and 150 mM KCl and spotted on nitrocellulose membranes (5 μL each) for the dot blot assay.

### Cytotoxicity assay of f-αSyn

For the visualization of cellular uptake of f-αSyn after fibrillization, PC12 cells were cultured on glass coverslips in 12-well plates at $3 \times 10^4$ cells/well with differentiation medium. At DIV 4, cells were treated for 24 h with 100 μL of each 7 μM f-αSyn solution after 18 h of incubation as prepared above (final 0.7 μM in culture medium), fixed with 4% PFA and immunostained using rabbit anti-FLAG antibody (1:200, Cell Signaling, 14793S), mouse Y136DOPAmab, Goat anti-Mouse IgG Secondary Antibody, Alexa Fluor 546 (1:1000, Invitrogen, A11003) and Goat anti-Rabbit IgG Secondary Antibody, Alexa Fluor 488 (1:1000), following the steps above.

For the cytotoxicity assay, an IncuCyte live-cell analysis system (Sartorius) was applied to monitor cell death. PC12 cells were cultured in 96-well plate at 6000 cells/well. At DIV 4–5, cells were treated with 5 μL of above f-αSyn solution (final 0.7 μM) or PBS and IncuCyte Cytotox Green Dye at a concentration of 200 nM. Then, the plate was placed in an IncuCyte incubator at 37 °C. Dead cells with green fluorescence were monitored and analyzed with IncuCyte ZOOM software according to the manufacturer's instructions. The data were obtained from distinct 3 wells per each sample at the same time.

### Sucrose density gradient centrifugation

The f-αSyn after fibrillization prepared above (7 μM, 400 μL each) were layered on a 12 mL 5–30% (w/v) linear sucrose gradient in 20 mM HEPES, pH 7.4, 150 mM KCl, 0.1% NP-40 and 0.5 mM DTT and centrifuged at 290,000 × g for 24 h. The tube was then punctured at the bottom, and 12 of 1-mL fractions were collected. The fractions were examined by WB using a mouse anti-αSyn antibody.

### Size exclusion chromatography (SEC)

The f-αSyn after fibrillization prepared above (7 μM, 100 μL each) or monomeric f-αSyn (7 μM, 100 μL each) were loaded onto a Superdex 200 Column 5/150 GL (Cytiva). SEC was performed using isocratic elution with SEC buffer (20 mM HEPES, pH 7.4, 150 mM KCl) at a flow rate of 1 mL/min on an AKTA explorer system (Cytiva). The eluted peaks were monitored at 280 nm.

### Structural models

Three-dimensional structures of rat TH (PDBID 1TOH) and human αSyn (PDBID 1XQ8) were obtained from the Protein Data Bank (PDB)[35,38]. Cavities on the TH (1TOH) surface were calculated by the program GHECOM (grid-based HECOMi finder; http://strcomp.protein.osaka-u.ac.jp/ghecom/) using the parameter of 10 Å radius for the large sphere probe[59] and visualized using Jmol, an open-source Java viewer for chemical structures in 3D (http://www.jmol.org/).

### Reporting summary

Further information on research design is available in the Nature Portfolio Reporting Summary linked to this article.

## Data availability

The data supporting the findings of this study are available in the manuscript and its supplementary information file. The TH and αSyn structure data used in this study are available in the Protein Data Bank under accession code PDB-1TOH and PDB-1XQ8, respectively. The statistics data generated in this study are provided in the Source data file. Source data are provided with this paper.

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

## Acknowledgements

We thank Prof. Kenji Iwasaki (Tsukuba University) and Dr. Fiona Francis (Sorbonne Universite) for providing valuable suggestions and critical comments, Ms. Miyuki Kira (Osaka Metropolitan University) and Ms. Yoriko Yabunaka (Osaka Metropolitan University) for technical supports of DNA sequencing and HPLC analysis, Mr. Hideki Nakagawa (Osaka Metropolitan University) for technical assistance with electron microscopy. We will also thank Mr. Hiromichi Nishimura (Osaka Metropolitan University) and Ms. Junko Hirohara (Osaka Metropolitan University) for mouse breeding. The result data were partially obtained in Research Support Platform, Osaka Metropolitan University Graduate School of Medicine. This work was supported by JSPS KAKENHI (JP17H04047 to S.H., JP21K06821 to M.J., JP19K16525 and JP22K06887 to S.M.) and Practical Research Project for Rare/Intractable Diseases from AMED (JP18ek0109390 to S.H., M.J., and S.M.). This research is also partially supported by the program for Moonshot R&D from JST (JPMJMS2024 to H.Y., T.Taguchi, and R.T.), the program for Brain/MINDS from AMED (JP19dm0207070 to R.T., JP20dm0207057 and JP21dm0207111 to H.H.), the Cooperative Research Program of Institute for Protein Research, Osaka University (CR-16-05 to S.H.) and JSPS KAKENHI (JP16H06276 (AdAMS) to S.H.).

## Author contributions

S.H. conceptualized and supervised the study, designed experiments and performed in vitro reaction, MS analysis, sucrose gradient centrifugation and SEC. M.J. performed fibrillization, dot blot assay, TEM and cytotoxicity analysis. S.M. performed antibody production, ELISA, solubility assay and viral overexpression analysis in cells and mouse, and made all figures. T.A. and N.T. performed immunohistochemical analysis using human sections. H.Y. and T.Taguchi generated Tg mouse and performed immunohistochemistry using its sections. T.Takao performed and supervised MS analysis. A.K. and H.H. generated AAV. R.T. provided supports for experimental design and data interpretation. S.K. provided a technique for the stereotaxic injection. R.I and S.C. contributed to collect experimental data. H.N. analyzed the cavity structure on TH surface. S.H., M.J., and S.M. wrote the initial manuscript, which was then revised and edited by all authors.

## Competing interests

The authors declare no competing interests.
