## [Peer Review File · Nature Communications]

DOPAnization of tyrosine in α -synuclein by tyrosine hydroxylase leads to the formation of oligomersREVIEWER COMMENTS

Reviewer #1 (Remarks to the Author):

In the study by Jin et al., the authors describe a posttranslational modification of alpha-Synuclein (aSyn) on tyrosine 136, resulting in its conversion to dihydroxyphenylalanine (Y136DOPA). Y136DOPA was identified *in vitro* using mass spectrometry. Furthermore, the authors developed a mouse monoclonal antibody, with which Y136DOPA was also identified in established cell and animal models for Parkinson's disease and in human brain samples. The authors further argue, that TH-induced dopanization leads to the formation of oligomeric species, that might be toxic, and that may seed further aggregation.

Overall, the idea that dopamine modifications might affect aSyn aggregation is not novel, and has been documented several times. What is novel is the mapping of this modification to Y136.

The authors point out that the brain areas affected in PD express tyrosine hydroxylase (TH) the rate limiting enzyme of catecholamine biosynthesis. The authors continue by briefly mentioning that TH hydroxylates tyrosine side chains, generating DOPA. The following hypothesis: "TH might participate in PD pathogenesis by modulating the intraprotein tyrosine residue of aSyn." seems a bit disconnected here. Authors should elaborate more on how this research hypothesis was initially generated.

The authors state that on page 3 "with wild-type aSyn and its two mutant forms identified in familial PD: aSyn E46K (E46K)¹¹ and aSyn A53T (A53T)⁴". However, several more aSyn mutations are known to cause familial forms of PD. The authors should clearly state this and also explain why the two mutants used here were chosen for detailed analysis. In general, showing the data for the aSyn mutation in the main figures does not add anything to the overall story of the paper.

Complete immunoblots should be shown for the new mAb antibody so the reader can assess the pattern of reactivity.

Some sentences need to be rephrased or written in a clearer manner. In line 55 "To explore the interaction between TH and aSyn" sentence is not precise because they are checking here if TH can hydroxylate Tyrosine residues in aSyn rather than its interaction with aSyn. In line 61-62 "We next digested these purified aSyns to make small fragments for mass spectrometry (MS)." the sentence is not clear what they mean by these purified aSyn and which peak they used. Lines 65-67 "The large fragment, residues 103-140, was subsequently digested by Asp N, theoretically making two fragments" what does theoretically mean? Add references for this.

Extended Figure 2a: Authors show RP-HPLC profiles of the tryptic digestion of TH-treated and control aSyn and mention that several new peaks are introduced upon TH reaction. In fact, the largest peak in the "control" spectra disappears more or less completely in the "reactant", suggesting an almost 100% conversion rate. Later, the authors mention a 40% rate for the conversion to aSyn Y136DOPA. How could this discrepancy be explained?

Figure 2d and extended Fig. 6b: Y136DOPAmAb seems to localize to the nucleus. However, mCherry seems to be localized to the cytoplasm, even though this is difficult to judge, since the mCherry channel seems overexposed. Better images should be included.

The authors should also use pS129 antibody staining to assess the phosphorylation status, as this is usually used as a pathology-associated modification.

The authors show that mCh-aSyn was dopanized at Tyr136 in cell and animal models. Was an alternative enzyme considered as potentially reacting with aSyn? Stating that "overexpressed by mCh-aSyn was dopanized [...] by endogenous TH *in vivo*" is maybe too confident, since the authors did not show direct interaction of aSyn and TH *in vivo*.

As the title of the paper already says, the authors argue that dopanization of aSyn leads to altered aggregation kinetics of aSyn. This statement is only based on TEM analyses and sucrose gradient centrifugation. TEM analyses revealed the presence of "insoluble large fibril aggregates" in control aSyn (however, the authors did not perform any solubility assay). Extended Fig. 7 shows that

Y136 aSyn does not form higher molecular weight species. Without further biochemical characterization of the modified protein, it is not justified to claim that oligomers were formed. The authors acknowledge this with saying that dopanized aSyn "tends" to form oligomers in some places. However, a number of established assays are available to analyze aSyn aggregation kinetics (ThT based aggregation assays, size exclusion chromatography, sequential extraction etc). If the authors point out in the title that oligomers are formed, this point needs to be more solid.

Figure 4: The aSyn concentration that was used to treat the PC12 cells is not mentioned anywhere. In the "methods" section it is only mentioned that 100uL of aSyn solution was added to the cells (in 12 well plates). Was it 5uM, as stated for the in vitro reaction of aSyn and TH? Was the aSyn solution somehow purified prior to adding to the cells? The authors say that fibrils/oligomers were uptaken. This is only based on the presence of FLAG and Y136DOPAmab signal inside the cells. However, as shown in the sucrose gradient centrifugation the aSyn solutions also contain monomeric aSyn. In general, any aSyn solution containing aggregated aSyn species also contains smaller species and monomeric protein because the aSyn species constantly convert into each other. Without purifying the aSyn species, the authors cannot know which aSyn species were internalized and thus, should not conclude that the dopanized aSyn oligomers are responsible for the observed cytotoxicity. Therefore, the text should be changed.

In immunoblotting and immunocytochemistry experiments, authors found aSyn to be dopanized. However, Quantification for the level of this new PTM was not done among different conditions. This would be nice to do especially to compare between the different aSyn mutants used in this study.

One last point is if TH can modify aSyn under normal physiological conditions. In the study, this modification is observed under overexpression or pathological conditions. A paragraph about this need to be written in the discussion part.

Minor points:

Line 158 160 references, references need to be added

The authors should discuss why tyr 133, 125 are not targets for TH hydroxylation. Another important point that needs discussion is whether TH hydroxylation is specific for aSyn or other proteins could be also targets.

Line 172 references from original articles need to be added

Extended data figure 1b : a short descriptive title on the top of the figure and also other figures is recommended. Extended data figure 1b, the absorbance values on Y-axis are missing and need to be present. I am wondering where is the peak for TH on the chromatogram. There is a small peak at 14 min, what this could be?

Extended data figure 2a.: The title of 2a and 2b need to be more accurate, it is RP-HPLC profiles of the digests not just trypsin digestion or Asp-N Digestion. in the control aSyn chromatogram it seems there is not only two peaks but also one more small peak close to 7.5 min, what is this peak.

Extended Data Fig. 1b: The second large peak is probably TH?

Line 68; „and separated by PR-HPLC“ A reference to the figure should be included here, not only in the following sentence.

Figure legend Extended Data Fig. 2 b: „fractions were applied to Asp-N digests“ should be corrected to „digestion“.

Several typos should be corrected throughout.

Line 79: "could not detect" instead of "could not detected"

Figure 2: "leNtivirus infection"

Figure legend 2c: „overexpressEing“

aSyn instead of aSyns – this should be corrected throughout the manuscript

„proteins were reacted“ and similar terms are not grammatically correct and should be revised throughout the manuscript

Reviewer #2 (Remarks to the Author):

The manuscript presents a strong case of involvement of tyrosine DOPAnization of α -synuclein in potential pathogenesis of Parkinson's disease. The authors have demonstrated the modified site in protein α -synuclein with some neatly designed experiments including the use of Y136DOPAmab for specificity. Despite the evidence provided, many details are missing, especially for the mass spectrometric data. Overall, the writing needs to be more comprehensible, interpretive, explanatory and as convincing as the experimental evidence itself. As of now, it is a little uneven and cursory for the reader. My questions/comments are listed below:

1. Please provide the missing LC gradients for the 2nd and 3rd separations and the LC column details for the 3rd separation.
2. The Raw MS data files provided do not open. T2d file format is not very commonly used, and I could not access those files. Please submit MS data in Microsoft Excel or another common format.
3. There are no details on the acquisition parameters of MS runs, how the data analysis and database search was performed and how the spectral figures were made. This is surprising since mass spec data is presented as the most important piece of evidence that points to the hydroxyl modification at the amino acid level.
4. Authors have convincingly demonstrated the non-possibility of Tyr 39, Tyr 133, Met 127 being the hydroxyl modified sites. However, Tyr 125 from the fragment DNEAY has not been discussed. It is mentioned in line 79 of the manuscript but the evidence in the figures indicated is missing.
5. Fig 1b- what is the dominant peak at around 745 m/z?
6. Fig 4c- what are the three curves (none, C, R)?

REVIEWER COMMENTS

Reviewer #1 (Remarks to the Author):

5 In the study by Jin et al., the authors describe a posttranslational modification of alpha-Synuclein (aSyn) on tyrosine 136, resulting in its conversion to dihydroxyphenylalanine (Y136DOPA). Y136DOPA was identified in vitro using mass spectrometry. Furthermore, the authors developed a mouse monoclonal antibody, with which Y136DOPA was also identified in established cell and animal models for Parkinson's
10 disease and in human brain samples. The authors further argue, that TH-induced dopanization leads to the formation of oligomeric species, that might be toxic, and that may seed further aggregation.

Overall, the idea that dopamine modifications might affect aSyn aggregation is not novel, and has been documented several times. What is novel is the mapping of this
15 modification to Y136.

We would like to thank this reviewer for providing critical comments and valuable suggestions. Concerning the reviewer's comments and to clarify, to the best of our knowledge, our manuscript is the first-time report to demonstrate that tyrosine
20 hydroxylase (TH) converts a tyrosine residue to DOPA within a protein and that its target is Tyr136 in α Syn. This is a newly identified posttranslational modification (PTM) by TH, not by dopamine as suggested by the reviewer. Our findings will provide new insight into the function of TH, because it has been recognized that the substrate of TH is free tyrosine molecules. This is the core of our study, which guides us to address
25 the mechanism by which abnormally accumulated α Syn selectively attacks dopaminergic neurons in Parkinson's disease (PD).

We performed additional experiments and rewrote the manuscript to address the criticisms from this reviewer as follows:

30 The authors point out that the brain areas affected in PD express tyrosine hydroxylase (TH) the rate limiting enzyme of catecholamine biosynthesis. The authors continue by briefly mentioning that TH hydroxylates tyrosine side chains, generating DOPA. The following hypothesis: "TH might participate in PD pathogenesis by modulating the

intraprotein tyrosine residue of aSyn.” seems a bit disconnected here. Authors should
35 elaborate more on how this research hypothesis was initially generated.

We appreciate this reviewer for a valuable suggestion and agree that our explanation
of how this hypothesis was raised was not sufficient. Although α Syn aggregation has
been well established as the cause of neurodegeneration, it was not enough to prove
40 the tropism of pathology in PD. We focused on the distribution of TH expression
overlapping with the impaired regions in PD, such as the substantia nigra. Many
studies have shown that α Syn functionally interacts with TH and regulates TH activity
and dopamine synthesis. However, whether TH enzyme activity affects the
characteristics of α Syn aggregation has not been reported. Therefore, we first
45 examined whether TH activity alters the conformation of α Syn aggregates using
HEK293 cells co-expressing α Syn with TH (Figure for reviewers 1). Surprisingly, the
formation of the thioflavin-S-positive β -sheet structures of α Syn aggregates was
suppressed depending on the TH enzyme activity (Figure for reviewers 1). This result
provided us with the hypothesis that TH activity is involved in the alteration of the
50 aggregation kinetics of α Syn as a PTM enzyme and contributes to the preferential loss
of dopaminergic neurons. To confirm our hypothesis, we designed the study presented
in this paper. In the revised manuscript, we have rewritten the text in the Introduction
section to elaborate upon the background of our hypothesis.

55 The authors state that on page 3 “with wild-type α Syn and its two mutant forms
identified in familial PD: α Syn E46K (E46K)¹¹ and α Syn A53T (A53T)⁴”. However,
several more α Syn mutations are known to cause familial forms of PD. The authors
should clearly state this and also explain why the two mutants used here were chosen
for detailed analysis. In general, showing the data for the α Syn mutation in the main
60 figures does not add anything to the overall story of the paper.

We agree that this point requires clarification. In this study, we used the familial PD
mutants E46K and A53T as well as wild-type α Syn to examine whether the
dopanization causes differences in fibrilization among wild-type and these mutant
65 forms. Previous works have reported that E46K and A53T mutations alter the fibril
structures and stabilize them (Li Y et al., 2018 *Cell Res.*, Zhao K et al., 2020 *Nat*

Commun.), leading to promotion of the formation of fibrillar aggregations. Furthermore, these mutants cause the conformational variants in their fibrils, which induces a clinicopathological heterogeneity of neurodegenerative diseases (Lau A et al., 2020 *Nat Neurosci*, Suzuki G et al., 2020 *Elife*). In spite of our expectation, we could not detect the clear differences in the conformation of aggregates within dopanized wild-type and two mutants. Y136DOPA modification induced the similar oligomer formation in all α Syn variants. This result suggests that dopanization may be an important PTM for promoting the oligomer formation of α Syn. We have rewritten the text regarding these mutants in the Results and Discussion sections to better illustrate the mutant data in terms of the overall story of the paper.

Complete immunoblots should be shown for the new mAb antibody so the reader can assess the pattern of reactivity.

80

We added all complete immunoblot images in Supplementary information file.

Some sentences need to be rephrased or written in a clearer manner. In line 55 " To explore the interaction between TH and α Syn" sentence is not precise because they are checking here if TH can hydroxylate Tyrosine residues in α Syn rather than its interaction with α Syn. In line 61-62 " We next digested these purified α Syns to make small fragments for mass spectrometry (MS)."the sentence is not clear what they mean by these purified α Syn and which peak they used. Lines 65-67 " The large fragment, residues 103-140, was subsequently digested by Asp N, theoretically making two fragments" what does theoretically mean? Add references for this.

90

We thank you for the detailed suggestions. We modified the text to improve the stylistic expression and added information about the α Syn fraction used for tryptic digestion in Supplementary Fig. 1b and its legend. In addition, we agree with this reviewer about the ambiguity of the word "theoretically". Although Asp-N normally cleaves the N-terminus of aspartic acid, it also cleaves that of Glu126 of α Syn atypically. In addition, through an additional experiment for this revision, we also found the previously missed cleavage of Asp121 of α Syn by Asp-N, collectively resulting in three fragments containing tyrosine. Therefore, we have removed "theoretically" from the main text and

95

100 added information about Asp-N cleavage to the figure legend of Fig. 1.

Extended Figure 2a: Authors show RP-HPLC profiles of the tryptic digestion of TH-treated and control α Syn and mention that several new peaks are introduced upon TH reaction. In fact, the largest peak in the “control” spectra disappears more or less completely in the “reactant”, suggesting an almost 100% conversion rate. Later, the authors mention a 40% rate for the conversion to α Syn Y136DOPA. How could this discrepancy be explained?

As this reviewer mentioned, the HPLC chromatogram of tryptic digests showed that the peak corresponding to 103-140 aa in control α Syn was decreased by more than 40% in that of reactant (Supplementary Fig. 2a). On the other hand, the Y136DOPA conversion rate assessed using full-length α Syn by ELISA was 40% (Supplementary Fig. 5c, d). By TH-mediated hydroxylation, a phenol on tyrosine is converted into a catechol, which tends to generate reactive oxygen species (ROS) by autoxidation. This ROS leads to the non-specific cleavage of the peptide bonds of proteins nonenzymatically. Based on this insight, we assumed that the “DOPA residue” in the C-terminal fragment may affect the stability of the peptide in trypsin digestion buffer. To address this notion, we newly prepared four α Syn oligopeptides, each of which contained Tyr125, Tyr133, DOPA125 or DOPA133, in addition to the Tyr136 and DOPA136 peptides made before. These peptides were incubated with or without trypsin digestion buffer (trypsin not included) for 16 h and examined by RP-HPLC. After incubation, the chromatograms showed that the peak of Tyr125 and Tyr133 peptides remained stable, whereas that of DOPA125 and DOPA133 peptides almost disappeared (Figure for reviewers 2a, b), suggesting the non-enzymatic degradation by the ROS released from DOPA residues. Interestingly, the peptide containing DOPA136 was somewhat more stable than the others (Figure for reviewers 2c). These results suggest that there is a possibility that dopanization at Tyr125 and Tyr133 may make the C-terminal fragment much less stable through the tryptic digestion process, resulting in the apparent > 40% reduction of the 103-140 aa fragment peak. In addition, these results may also provide the reason why we did not detect the dopanized Tyr125 and Tyr133 by MS in this study.

Figure 2d and extended Fig. 6b: Y136DOPAmab seems to localize to the nucleus. However, mCherry seems to be localized to the cytoplasm, even though this is difficult
135 to judge, since the mCherry channel seems overexposed. Better images should be included.

Previous work has already revealed that α Syn is localized in presynaptic terminals, the nucleus and the cytoplasm (Goers J et al., Biochemistry. 2003, Yu S et al.,
140 Neuroscience. 2007). Consistently, mCherry- α Syn was observed in both the nucleus and cytoplasm of PC12 cells in our study. However, we agree with the point that mCherry fluorescence was overexposed. We replaced former images with new images in Fig. 2d and Supplementary Fig. 6b.

145 The authors should also use pS129 antibody staining to assess the phosphorylation status, as this is usually used as a pathology-associated modification.

In accordance with this reviewer's comment, we assessed the phosphorylation status of mCh- α Syn in PC12 cells using the pS129 antibody. Immunocytochemistry was used
150 to detect the pS129 phosphorylation signal overlapping with dopanized mCh- α Syn, suggesting that dopanization may be involved in the initiation or progression of α Syn pathology. We added these data to Supplementary Fig. 6h.

The authors show that mCh-aSyn was dopanized at Tyr136 in cell and animal models.
155 Was an alternative enzyme considered as potentially reacting with aSyn? Stating that "overexpressed by mCh-aSyn was dopanized [...] by endogenous TH *in vivo*" is maybe too confident, since the authors did not show direct interaction of aSyn and TH *in vivo*.

We appreciate this important suggestion. The direct interaction of α Syn and TH *in vivo*
160 has been reported, in which α Syn regulates dopamine metabolism by interacting with TH (Pifl C et al., Neurosci Lett. 2004). To address whether TH is responsible for the dopanization of mCh- α Syn *in vivo*, we evaluated the Y136DOPA modification of mCh- α Syn in PC12 cells after treatment with the TH inhibitor, α -methyl tyrosine (AMPT). Western blot analysis showed that the dopanization of exogenous mCh- α Syn was

165 clearly decreased after AMPT treatment. This result indicates that the enzymatic activity of endogenous TH directly contributes to the Y136DOPA modification of mCh- α Syn. We added this result to Fig. 2g and Supplementary Fig. 6g.

As the title of the paper already says, the authors argue that dopanization of aSyn leads to altered aggregation kinetics of aSyn. This statement is only based on TEM analyses and sucrose gradient centrifugation. TEM analyses revealed the presence of “insoluble large fibril aggregates” in control aSyn (however, the authors did not perform any solubility assay). Extended Fig. 7 shows that Y136 aSyn does not form higher molecular weight species. Without further biochemical characterization of the modified protein, it is not justified to claim that oligomers were formed. The authors acknowledge this with saying that dopanized aSyn “tends” to form oligomers in some places. However, a number of established assays are available to analyze aSyn aggregation kinetics (ThT based aggregation assays, size exclusion chromatography, sequential extraction etc). If the authors point out in the title that oligomers are formed, this point needs to be more solid.

Thank you very much for this valuable suggestion. In accordance with this reviewer’s comment, we first performed a detergent solubility assay of FLAG- α Syn (f- α Syn) aggregates and analyzed them by dot blot assay using a Syn-O2 antibody. After f- α Syn was incubated for 18 h, the sarkosyl-insoluble aggregates were considerably increased in control f- α Syn compared with those in dopanized f- α Syn (Supplementary Fig. 7b). However, these data do not directly indicate whether the aggregates in the TEM images of control f- α Syn are insoluble. Therefore, we removed the word “insoluble” from the Abstract and Results section and added the solubility assay data in Supplementary Fig. 7b and modified the text in the Results section.

We next performed size exclusion chromatography (SEC) using dopanized f- α Syn after 18 h of incubation. Dopanized f- α Syn exhibited the formation of oligomers up to the 15-mers (Fig. 4c, Supplementary Fig. 7e). This is consistent with the results from TEM analysis and sucrose gradient centrifugation. We added the SEC data in Fig. 4c and Supplementary Fig. 7e and modified the text in the Results section.

Figure 4: The aSyn concentration that was used to treat the PC12 cells is not mentioned anywhere. In the “methods” section it is only mentioned that 100uL of aSyn solution was added to the cells (in 12 well plates). Was it 5uM, as stated for the in vitro reaction of aSyn and TH? Was the aSyn solution somehow purified prior to adding to the cells? The authors say that fibrils/oligomers were uptaken. This is only based on the presence of FLAG and Y136DOPAmab signal inside the cells. However, as shown in the sucrose gradient centrifugation the aSyn solutions also contain monomeric aSyn. In general, any aSyn solution containing aggregated aSyn species also contains smaller species and monomeric protein because the aSyn species constantly convert into each other. Without purifying the aSyn species, the authors cannot know which aSyn species were internalized and thus, should not conclude that the dopanized aSyn oligomers are responsible for the observed cytotoxicity. Therefore, the text should be changed.

We prepared fibrillized f- α Syn samples at a 7 μ M concentration as described in the paragraph “Fibrillization of f- α Syn and TEM” in the Methods section and used these samples to treat PC12 cells (Fig. 4d and Supplementary Fig. 8a). We added information about the final concentration of each fibril solution in the culture medium to the Methods section.

Furthermore, we agree with the second part of the reviewer’s comment. The aggregated samples of control and dopanized f- α Syn used for cell treatment also contain monomeric species, and the α Syn species constantly convert into each other. We have changed the text throughout this paper carefully in accordance with the comment from this reviewer.

In immunoblotting and immunocytochemistry experiments , authors found aSyn to be dopanized. However, Quantification for the level of this new PTM was not done among different conditions. This would be nice to do especially to compare between the different aSyn mutants used in this study.

In accordance with this reviewer’s comment, we examined the level of Y136DOPA modification in the overexpressed mCh- α Syn in PC12 cells by chemiluminescence ELISA. We confirmed that the Y136DOPA conversion rate of mCh- α Syn was 4-5%,

which is not obviously different among mCh- α Syn, mCh-E46K and mCh-A53T (Figure for reviewers 3).

235 One last point is if TH can modify α Syn under normal physiological conditions. In the study, this modification is observed under overexpression or pathological conditions. A paragraph about this need to be written in the discussion part.

We modified the text in the Discussion section, according to the request from this reviewer.

240

Minor points:

Line 158 160 references, references need to be added

We have added the references.

245

The authors should discuss why tyr 133, 125 are not targets for TH hydroxylation.

250 We could not detect clear dopanization at Tyr133 and Tyr125 in this study. However, the additional experiment of RP-HPLC using DOPA125 and DOPA133 peptides described above revealed their instabilities during our experimental operations (Figure for reviewers 2), which might make MS analysis extremely challenging. Accordingly, there are two possible explanations for this: (1) Tyr125 and 133 are also dopanized, but these DOPA-containing peptides are degraded during the experiment; (2) there may be a consensus sequence for dopanization that excludes Tyr125 and 133 from the target of TH. We added the text about these considerations to the Discussion section.

255

Another important point that needs discussion is whether TH hydroxylation is specific for α Syn or other proteins could be also targets.

260

This is a very interesting suggestion for us. In this study, α Syn dopanization was identified at the C-terminal region, which is an intrinsically disordered domain (IDD) lacking a tertiary structure. IDDs have been reported to be susceptible to PTMs, which

leads to alterations in protein conformation. Therefore, we assumed that TH
265 hydroxylation may target IDD in other proteins and tried examining the dopanization
of tau. Tau is one of pathogenic proteins of tauopathy that contains a several diseases
leading to a preferential loss of dopaminergic neurons like PD. MS analysis showed a
possibility that TH dopanized the recombinant human tau at Tyr197 and Tyr310 (Figure
for reviewers 4). Although our preliminary data suggest that dopanization may not be
270 specific for α Syn, further experiments are necessary to prove the generality of this PTM.
In the revised manuscript, we added the text about the possibility of the dopanization
of other proteins with IDDs in the Discussion section.

Line 172 references from original articles need to be added

275

We have exchanged for an original article.

Extended data figure 1b : a short descriptive title on the top of the figure and also other
figures is recommended. Extended data figure 1b, the absorbance values on Y-axis
280 are missing and need to be present. I am wondering where is the peak for TH on the
chromatogram. There is a small peak at 14 min, what this could be?

We added a short title and Y-axis value to the Supplementary Fig. 1b. In this
experiment, we used GST-tagged α Syn for *in vitro* reaction with TH and separated
285 them from the reaction mixture by using GST Sepharose and PreScission protease
(for GST removal) before RP-HPLC (described in the Methods section). Thus, the peak
for TH does not appear in this chromatogram. The small peak at 14 min corresponds
to free GST contaminants contained in the samples after PreScission protease
cleavage. We added the detailed experimental procedures to the figure legend of
290 Supplementary Fig. 1b.

Extended data figure 2a.: The title of 2a and 2b need to be more accurate, it is RP-
HPLC profiles of the digests not just trypsin digestion or Asp-N Digestion. in the control
aSyn chromatogram it seems there is not only two peaks but also one more small peak
295 close to 7.5 min, what is this peak.

We have corrected the short title of Supplementary Fig. 2a and b. As this reviewer mentioned, the chromatograms for the control showed an extra small peak close to 7.5 min after trypsin digestion. This peak does not correspond to the tyrosine-containing peptide derived from the tryptic digestion of α Syn. At OD280 nm, the phenylalanine (Phe) residue also has a certain absorption, but it appears as a much weaker peak than that of tyrosine. Thus, it is possible that this small HPLC peak near 7.5 min may correspond to the α Syn fragment containing Phe (1-6 or 81-96 aa) or to a trypsin fragment caused by its autodigestion.

305

Extended Data Fig. 1b: The second large peak is probably TH?

TH was removed from the *in vitro* reaction mixture before RP-HPLC separation as described above (in the response to the above comment). Both of the two peaks at 18-19 min in the reactant chromatogram corresponds to α Syn, thus we collected the fraction containing these peaks for the subsequent tryptic digestion. Furthermore, the large peak at 22 min were caused by detergent contained in the samples. We have added information about α Syn peaks to Supplementary Fig. 1b and its legend.

310

Line 68; „and separated by PR-HPLC“ A reference to the figure should be included here, not only in the following sentence.

Figure legend Extended Data Fig. 2 b: „fractions were applied to Asp-N digests“ should be corrected to „digestion“.

315

We modified the text according to reviewer's comment.

320

Several typos should be corrected throughout.

Line 79: “could not detect” instead of “could not detected”

Figure 2: “leNtivirus infection”

325

Figure legend 2c: „overexpressEing“

aSyn instead of aSyns – this should be corrected throughout the manuscript

„proteins were reacted“ and similar terms are not grammatically correct and should be revised throughout the manuscript

330 We corrected these typing errors and stylistic expressions through the Main text, figures and figure legends according to reviewer's comment.

335 Reviewer #2 (Remarks to the Author):

The manuscript presents a strong case of involvement of tyrosine DOPAnization of α -synuclein in potential pathogenesis of Parkinson's disease. The authors have demonstrated the modified site in protein α -synuclein with some neatly designed experiments including the use of Y136DOPAmab for specificity. Despite the evidence provided, many details are missing, especially for the mass spectrometric data. Overall, the writing needs to be more comprehensible, interpretive, explanatory and as convincing as the experimental evidence itself. As of now, it is a little uneven and cursory for the reader.

345

We thank this reviewer for valuable suggestions and critical comments. In accordance with this reviewer's comment, we modified the text as follows:

My questions/comments are listed below:

350 1. Please provide the missing LC gradients for the 2nd and 3rd separations and the LC column details for the 3rd separation.

We have now added requested information for the LC gradients and column to the Methods section.

355

2. The Raw MS data files provided do not open. T2d file format is not very commonly used, and I could not access those files. Please submit MS data in Microsoft Excel or another common format.

360 We apologize for the inconvenience caused by our MS data file's format. We have supplemented Raw MS data in CSV and Microsoft Excel format.

3. There are no details on the acquisition parameters of MS runs, how the data analysis and database search was performed and how the spectral figures were made. This is
365 surprising since mass spec data is presented as the most important piece of evidence that points to the hydroxyl modification at the amino acid level.

We recognize the importance of the reviewer's suggestion for this point. We modified the text by adding detailed information of the MS analysis in the Methods section. In
370 this study, we did not use the database search, since the peptide mass of trypsin and/or Asp-N digests of recombinant α Syn was predicted from its amino acid sequence.

4. Authors have convincingly demonstrated the non-possibility of Tyr 39, Tyr 133, Met 127 being the hydroxyl modified sites. However, Tyr 125 from the fragment DNEAY
375 has not been discussed. It is mentioned in line 79 of the manuscript but the evidence in the figures indicated is missing.

We agree with the reviewer's comment on this point. We were not previously able to detect the MS peak corresponding to the DNEAY (121-125 aa) fragment of α Syn. Thus,
380 in an additional experiment, we synthesized an oligopeptide of α Syn spanning Tyr125 and digested it with Asp-N. Importantly, we found the previously missed cleavage by Asp-N at the N-terminus of Asp121 under our experimental conditions. For that reason, we reexamined the former MS data to search for the MS peak corresponding to DPDNEAY (119-125 aa) and found it (m/z 823.4). However, there was no obvious
385 peak that indicated the dopanization of Tyr125 (m/z 839.4). We added these data to Supplementary Fig. 3b and modified the text.

5. Fig 1b- what is the dominant peak at around 745 m/z ?

390 In accordance with the reviewer's comment, we performed MS/MS analysis of the peak at m/z 745. However, we could not identify the amino acid sequence.

6. Fig 4c- what are the three curves (none, C, R)?

395 None and C indicate the samples treated with PBS and control f- α Syn, respectively. In the revised manuscript, we exchanged 'R' for 'TH+' that indicates the samples treated with dopanized f- α Syn. We added the text about these symbols to the figure legends of Fig. 4e and Supplementary Fig. 8b.

REVIEWERS' COMMENTS

Reviewer #1 (Remarks to the Author):

The authors have significantly revised and strengthened the manuscript, so I have no further comments.

Reviewer #2 (Remarks to the Author):

Thank you for addressing my comments. Even though the data is now provided in the MS Excel format, the two columns with numerical data are not defined in any of the files. So, it is not clear what those two columns represent and how this data has been used in making figures. Also, it is still a mystery that the base peak at 745 m/z is unidentified.

REVIEWERS' COMMENTS

Reviewer #2 (Remarks to the Author):

Thank you for addressing my comments. Even though the data is now provided in the MS Excel format, the two columns with numerical data are not defined in any of the files. So, it is not clear what those two columns represent and how this data has been used in making figures. Also, it is still a mystery that the base peak at 745 m/z is unidentified.

We appreciate this reviewer for important comments, and again apologize for missing definition of the two columns in MS Excel files. In each file, the first column shows molecular weight (m/z) (X-axis of MS figures in the manuscript), and the second column shows intensity of MS peaks (Y-axis of MS figures that was indicated by % in the manuscript). We processed and visualized MS raw data through Data Explorer software (version 4.8; Applied Biosystems) as described in the Methods section, and exported the images of MS spectrum of interest to PDF files for making figures.

In terms of the base peak at 745 m/z in Fig. 1c, we have now identified it as the Na⁺-bound form of the corresponding peptide with 723 m/z. Mass spectra of peptides is artificially influenced by trace Na⁺ or K⁺ ion mainly derived from a stock glass bottle of the solvent. With MALDI-MS, protonated molecule ion [M+H]⁺ are normally observed. However, when trace amounts of alkali metal ions form clusters (adducts) with peptides, sodiated or potassiated molecule ions [M+Na/K]⁺ are also produced. In particular, this 723 m/z peptide is highly acidic, which is prone to make alkali metal ion adducts. Indeed, we also found the MS peak corresponding K⁺-bound form at 761 m/z. We show the MS spectrum of control αSyn used for Fig. 1c below (Figure for reviewer), and provided all PDF images used for making figures with the MS Excel raw data.

Figure for reviewer: The MS spectrum of Fig. 1c (control) in the manuscript. The original PDF image is provided in the MS raw data folder with the revised manuscript (file name: WT_control723.pdf).